# Mayo Clinic Clival Chordoma Case Series: Impact of Endoscopic Training on Clinical Care

**DOI:** 10.3390/cancers14205104

**Published:** 2022-10-18

**Authors:** Sukwoo Hong, Nadia Laack, Anita Mahajan, Erin K. O’Brien, Janalee K. Stokken, Jeffrey R. Janus, Garret Choby, Jamie J. Van Gompel

**Affiliations:** 1Department of Neurological Surgery, Mayo Clinic, Rochester, MN 55902, USA; 2Department of Radiation Oncology, Mayo Clinic, Rochester, MN 55902, USA; 3Department of Otolaryngology, Mayo Clinic, Rochester, MN 55902, USA

**Keywords:** chordoma, multidisciplinary, endoscopic, skull base, clivus, adjuvant radiation, proton

## Abstract

**Simple Summary:**

Clival chordoma is a rare malignant tumor that commonly arises near the center of the brain. In 2013, multidisciplinary team was introduced in our facility to manage clival chordoma effectively. We analyzed its effect by comparing the outcomes of patients from “before 2013” (19 patients) and “after 2013” (39 patients). After 2013, the endoscopic endonasal approach was more commonly used (90%; “before 2013”, 32%) to achieve gross to near total resections (64%; “before 2013”, 16%) with less post-operative new cranial nerve deficits (3%; “before 2013”, 32%). Gross to near total resection was associated with less recurrence (hazard ratio, 0.31; 95% confidence interval, 0.11 to 0.91; *p* = 0.033). The multidisciplinary team led to improved tumor resection rate, less tumor recurrence rate, and less morbidity.

**Abstract:**

The management of clival chordoma in our group shifted around 2013 to mostly endoscopic, and proton beam was introduced for our multidisciplinary team. Consecutive patients who had surgical resection from 1987 to 2021 were reviewed. A total of 58 patients (39 patients after 2013) were analyzed. The mean tumor size was 3.7 cm, and the most common location was the upper clivus (43%). Compared to before 2013, after 2013, the endoscopic endonasal approach was more common (90%, *p* < 0.001), and more gross or near total resections (64%, *p* = 0.002) were attained. Ten cases (17%) were revision surgeries referred from elsewhere, and three cases (5%) underwent additional surgery elsewhere before adjuvant radiation. The postoperative cerebrospinal fluid leak occurred in 7%. Post-operative new cranial nerve deficits occurred in 32% before 2013, compared to 2.6% after 2013 (*p* = 0.004). For cases before 2013, 10 patients (53%) recurred during the median follow-up of 144 months (mean, 142 months), whereas for cases after 2013, seven patients (18%) recurred with a median follow-up of 35 months (mean, 42 months). 5-year progression-free survival was 58%, and 5-year overall survival was 87%. A specialized multidisciplinary team improved the resection rate compared to a historical cohort with an excellent morbidity profile.

## 1. Introduction

Chordoma is a tumor, which originates from the notochordal remnants, and arises in the spinal axis from the skull base (32%) to the sacrum (29%) [1,2]. Its incidence is only 8 in 10,000,000 [1]. Chordoma is considered malignant, histologically and clinically, because it tends to invade surrounding bone and soft tissues resulting in multiple recurrences [2]. Further, it can metastasize, however infrequently to the skin and bone and less commonly to the lungs, lymph nodes, and heart [3,4]. Skull base chordoma most often involves the clivus. Due to the central and deep location, the approach to these lesions can be challenging, and tumor resection without damaging surrounding cranial nerves and vasculature is a major challenge. However, with the advent of endoscopy in the last two decades, endoscopic teams have developed progressively to deal with this formidable lesion. As a result, most clival chordomas are now resected endoscopically [5,6,7].

Prior to 2013, endoscopic approaches were present at our institution. However, recruitment brought in trained specialists to treat chordoma, and concomitant with this, proton beam therapy was built at our center to develop our multidisciplinary team further. Here, to understand the impact of specified endoscopic training at our facility, we compared this group’s outcome (from 2013 to 2021; “after 2013”) to the epoch prior to its introduction (from 1987 to 2012; “before 2013”).

## 2. Materials and Methods

The approval from the institutional review board was obtained (ID: 13-008054). The consecutive patients who underwent surgical resection in our facility from 1987 to 2021 for clival chordoma were identified. Biopsy was not considered a surgical resection. Patients’ demographic, clinical, radiological, and pathological data were retrospectively collected. The period between 1987 and 2021 was split in 2013 when the multidisciplinary team was recruited. The patients were classified based on the first surgery date at our facility. Overall survival (OS) and progression-free survival (PFS) were calculated from the completion of the radiation therapy or the date of surgery (for those who did not undergo adjuvant radiation). Tumor size was defined as the longest diameter based on magnetic resonance imaging (MRI) before the first surgery. For revision surgeries, the tumor sizes recorded by outside hospitals were used. The original tumor size was used for the recurrent case after hypofractionated stereotactic radiosurgery at an outside hospital. Tumors were classified into four based on location in the clivus: upper clivus, middle clivus, lower clivus, and holoclivus. As for lateral extension, the upper clivus included the cavernous sinus. The middle clivus included the petrous bone (the petroclival fissure). Furthermore, the lower clivus included the jugular foramen and the occipital condyle [8]. The degree of surgical resection was defined based on postoperative MRI as follows: the gross total resection (GTR) was defined as no residual enhancement, the near total resection (NTR) was defined as a thin rim of enhancement, and the subtotal resection (STR) was defined as the rest. Adjuvant radiation was done by stereotactic radiosurgery (Gamma Knife), external beam radiation therapy, or proton beam therapy.

The Shapiro–Wilk test was used to tell parametric and non-parametric data. Parametric data were written in mean ± standard deviation, whereas non-parametric data were written in the median with interquartile range (IQR). The variables were compared between the two periods by student’s *t*-test, Chi-squared test, or Fisher’s exact test. The postoperative complications and outcomes were evaluated with binary logistic regression to calculate the odds ratio (OR) and Cox regression to calculate the hazard ratio (HR). Univariate analysis was performed, followed by multivariate analysis. The variables with a *p*-value ≤ 0.05 were included in the multivariate analysis. Pearson coefficient of 0.7 was set as the cutoff to prevent multicollinearity in the multivariate analysis. Kaplan–Meier method was used to evaluate PFS and OS. The analysis was done using SPSS version 25.0 (IBM Inc., Armonk, NY, USA). A *p*-value ≤ 0.05 was considered statistically significant. 

## 3. Results

Sixty patients were identified. Two patients were excluded because they had a biopsy and did not have a surgery (a total of 58 patients). A summary of their baseline characteristics and treatment outcomes is summarized in Table 1. Nineteen patients were treated before 2013, representing a rate of 0.7 cases per year, from 1987 to 2012 (26 years). Two patients were referred from outside hospitals after a biopsy. There were no revision surgery cases during this period. From 2013 to 2021 (9 years), 39 patients were treated (4.3 cases per year). Twelve patients were referred from outside hospitals with pathological diagnoses. Among them, one patient had a biopsy, ten patients (26%) had surgeries, and one patient was treated with upfront hypofractionated stereotactic radiosurgery. Five cases were recurrent. In Table 1, previous treatment means surgeries or stereotactic radiosurgery. Revision surgery is included in previous treatment. Recurrent case means chordoma regrowth after the initial surgeries in previous hospitals and referred to us for further treatment. Recurrent case is included in previous treatment. As for the tumor location, the upper clivus (25 of 58 [43%]) was the most common site in both periods, followed by the middle clivus, holoclivus, and lower clivus (seven of 58 [13%]).

Before 2013, the endoscopic endonasal approach (EEA) was used in six of 19 cases (32%), and the rest were managed without an endoscope. After 2013, 30 of 39 cases (77%) were performed only by EEA, and 36 cases (92%) were performed using an endoscope (Figure 1a). The endoscopic high transcervical approach was chosen in one case for a residual chordoma in the peri-odontoid space after two EEAs and occipito-cervical fusion in an outside hospital. Five cases were a combination of EEA and other approaches. EEA followed by an open tumor resection with craniotomy was done in two cases where the tumors were hard to dissect safely from the surrounding structures due to scar tissue from prior operations at outside institutions. Occipito-cervical fusion was performed in three patients: two cases of lower clival chordoma, which involved the occipito-cervical junction, and one of holoclival chordoma, which involved the left occipital condyle. An occipito-cervical fusion after EEA (two stages) was performed in one patient, who was complicated by incision drainage, which was resolved with additional stitches and antibiotics. An occipito-cervical fusion before EEA (two stages) was performed in another patient. This patient required additional surgery via an endoscopic high transcervical approach to resect residual tumor near the odontoid process. The third patient with holoclival chordoma underwent occipito-cervical fusion before EEA (two stages). The latter two patients had uncomplicated postoperative courses without wound problems. Open craniotomy was used in 12 patients with or without EEA. Subtemporal craniotomy was most commonly utilized (four patients) followed by frontotemporal craniotomy (three patients), suboccipital craniotomy (two patients), temporal occipital craniotomy (two patients), and far lateral supracondylar craniotomy (one patient). 

After 2013, the rate of GTR/NTR improved from 16% (3 of 19 cases) to 64% (25 of 39 cases; *p* < 0.001 by Chi-squared test; Table 1; Figure 1b). Out of 30 cases performed only by EEA after 2013, GTR/NTR was attained in 22 cases (73%). The operative approach and the resection degree based on the four clival chordoma classification is summarized in Table 2. After 2013, the middle clival chordoma cases were all resected by EEA. Furthermore, GTR was accomplished in 80%, which was higher than the other clival regions. The location of the tumor residual in the STR group is summarized in Table 1. After EEA, the most common location was posterior to the paraclival internal carotid artery, the petrous apex, and the cavernous sinus. After open craniotomy, the residual tumor location varied depending on the craniotomy sites. After subtemporal craniotomy, the residuals ranged from the prepontine cistern (one patient), the cavernous sinus/petrous apex/cerebellopontine angle (one patient), to the sella/suprasellar regions (one patient). After temporal occipital craniotomy, all the residuals were in the prepontine cistern (two patients). After suboccipital craniotomy, the residual was in the cavernous sinus to medial to the petrous segment internal carotid artery (one patient).

### 3.1. Pathological Information

Data on chordoma subtypes [9] were Histological data were available in 10 patients (17%; five conventional chordomas and five chondroid chordomas); however, all patients had pathologic confirmation of chordoma. No patients had de-differentiated chordomas. The data on brachyury was available in 30 patients. Twenty-nine (97%) specimens showed positivity for brachyury. One showed positive brachyury in the biopsy specimen from an outside hospital, but a subsequent surgical specimen in our facility showed negative for brachyury. Chromosomal microarray analysis was done in two recurrent cases after surgeries in outside hospitals. Both had chromosome 1p loss/1q gain. One mutation was in the phosphatase and tensin homolog (PTEN) gene, and the other was in the tumor protein 53 (TP53) gene.

### 3.2. Surgical Complications

Postoperative new cranial nerve deficits decreased from 32% (6 of 19 cases) to 3% (1 of 39 cases; *p* = 0.004; Table 1). Binary logistic regression analysis for postoperative new cranial nerve deficits showed that patients treated before 2013 had an OR of 17 (95% confidence interval [CI], 1.9 to 160; *p* = 0.001). No other pertinent factors were significantly associated with postoperative new cranial nerve deficits (Table 3).

The rate of postoperative cerebrospinal fluid (CSF) leak increased from 0% to 10% (4 of 39 cases; *p* = 0.29; Table 1). Out of 30 surgical cases performed only by EEA after 2013, intraoperative CSF leak was observed in 23 (77%), of which six were low-flow CSF leaks, and 17 were high-flow CSF leaks. “Low-flow” CSF and “high-flow” CSF leaks were determined by primary surgeons’ impressions written in the operative notes. For dural reconstruction, the fascia lata was used in one of six low-flow CSF leaks and 11 of 17 high-flow CSF leaks. The nasoseptal flap was used in 24 patients (five of seven no CSF leaks; six of six low-flow CSF leaks; 13 of 17 high-flow CSF leaks). The gasket seal technique [10] was not used in any cases. Intraoperative CSF leaks were observed in all four patients, which resulted in postoperative CSF leaks. One of the four patients had a low-flow CSF leak, and the fascia lata was not used for duraplasty, and postoperative CSF leak had to be managed by endoscopic repair. The other three patients had high-flow CSF leaks, where the fascia lata was used in one (recovered with lumbar drainage) and not used in the other two (one required endoscopic repair; one recovered with lumbar drainage). A lumbar drain was placed prophylactically at the time of EEA in 20 patients (67%).

### 3.3. Adjuvant Radiation Therapy

Postoperatively, three cases underwent additional surgeries elsewhere (Table 1). After 2013, 35 patients (90%) received adjuvant radiation, compared to 12 patients (63%) before 2013 (*p* = 0.03). Before 2013, five patients (42%) received proton beam therapy, whereas, after 2013, all adjuvant radiation was done with proton beam therapy (*p* < 0.001; Figure 1c). The median dose and fractionation of proton beam therapy were 73.5 Gy (IQR, 70.0 to 75.9 Gy) and 37 (IQR, 35 to 38), respectively. The interval between the last day of surgery to the start of proton beam therapy was 80 days in median (IQR, 56 to 112 days). On the other hand, the mean dose of marginal dose and maximum dose of stereotactic radiosurgery were 14 Gy (±3 Gy) and 31 Gy (±6 Gy), respectively. As for external beam radiation therapy, the mean dose was 53 Gy (±4 Gy).

### 3.4. Outcome

During follow-up, 10 of 19 patients (53%) from before 2013 had a recurrence, whereas seven of 39 patients (18%) from after 2013 experienced a recurrence (*p* = 0.005; Table 1). The recurrence in the first six years after treatment showed no significant difference between the two groups (Figure 1d). For patients before 2013, the mean PFS and OS were 88 (±67) months and 149 (±76) months, respectively. The mean follow-up period was 142 months (median, 144 months). For patients after 2013, the median PFS and OS were 29 months (IQR, 11 to 51 months) and 35 months (IQR, 14 to 57 months), respectively. The mean follow-up period was 42 months (median, 35 months). During these periods, nine of 19 patients (47%) from “before 2013” died, whereas two of 39 patients (5%) from “after 2013” died. Univariate analysis on a recurrence by binary logistic regression showed that treatment group before 2013 (OR, 5.5; 95% CI, 1.6 to 19; *p* = 0.007), subtotal resection (OR, 3.8; 95% CI, 1.1 to 13; *p* = 0.03), and proton beam therapy (OR, 0.13; 95% CI, 0.026 to 0.61; *p* = 0.01) to be associated with a recurrence. However, multivariate analysis did not show any of them to be significant (Table 4). Cox regression analysis on a recurrence showed subtotal resection (HR, 3.2; 95% CI, 1.1 to 9.4; *p* = 0.03) as a prognostic factor (Table 5). The PFS rate and OS rate are summarized in Table 6. The 10-year PFS and OS rates could not be calculated for patients treated after 2013 because none of the patients were followed for ten years. PFS and OS were not significantly different before 2013 and after 2013 (Figure 2a,b). Patients for whom GTR/NTR were attained had better PFS (Figure 2c). Conversely, no significant difference in PFS was observed between the status of adjuvant radiation (Figure 2d).

## 4. Discussion

With our multidisciplinary team, we performed a significant proportion (64% [25 of 39]) of GTR/NTR after 2013. The average number of surgeries increased substantially from 0.7 to 4.3 cases yearly. Patients experienced fewer postoperative new cranial nerve deficits. Fewer patients have had recurrence during the median follow-up of 38 months (IQR, 14 to 63 months) (Table 1).

The preoperative patient or tumor characteristics did not change significantly between the two time periods, except for the proportion of revision surgeries and previous treatment in outside hospitals. The increased number of revision surgeries and referrals may reflect physicians’ recognition that clival chordoma needs multidisciplinary management. The most common clival chordoma location was the upper clivus (Table 1), which is in line with a previous study [6]. Most lesions were in the upper to middle clivus (39 of 58 patients [67%]), as was the case with a previous study (45 of 80 cases [56%]) [7]. We attained more GTR/NTR with EEA (22 of 30 [73%]), surgical techniques, and knowledge advancement. As can be seen from fewer postoperative new cranial nerve deficits (1 of 39 [3%]; *p* = 0.004; Table 1), surgical morbidity has lessened significantly. The same trend can be seen in a large study [6]. CSF leak is the most common surgical complication and can be the main drawback of EEA [6,11]. In our cohort, CSF leaks occurred more often after 2013 (*p* = 0.29). Although we could not detect a significant factor for postoperative CSF leak, using the fascia lata for duraplasty should be considered, mainly when intraoperative high-flow CSF leak was observed. Furthermore, the gasket seal technique where fascia lata is stabilized by a rigid buttress may reinforce the water tightness of the skull base defect. In one study, where the authors performed the gasket seal technique on 46 patients comprised of craniopharyngioma, meningioma, and pituitary adenoma, only two patients (4%) had a postoperative CSF leak [10]. In another study, CSF leaks were associated with large tumor volume and level of tumor invasiveness [12]. One meta-analysis on post-operative CSF leak risk factors after pituitary adenoma transsphenoidal surgery showed body mass index, tumor size, revision surgery, and intraoperative CSF leak as significant factors [13]. However, none of these factors were significant in our cohort. This could be due to some differences in dural reconstruction, lumbar drainage, and tumor characteristics. According to past studies, the risk of CSF leak after EEA or trans-sphenoidal surgery ranges from 0 to 35% [6,12,14,15,16,17]. In our study, the risk of CSF leak after EEA was 11% (four of 36), which falls into this range.

Although our study showed an improved tumor resection rate with less morbidity with the multidisciplinary team introduced in 2013, we could not show that this multidisciplinary approach was associated with less recurrence (Table 5). We believe this is due to the small number of cases and not long-enough follow-up. The only factor associated with less recurrence was the degree of resection. This result is consistent with other studies, which showed that EEA was associated with increased resection rate with low morbidity [6,7,18]. A meta-analysis on prognostic factors in skull base chordoma showed that STR (pooled HR, 2.0; 95% CI, 1.5 to 2.6) and adjuvant radiation (pooled HR, 0.30; 95% CI, 0.16 to 0.56) were associated with PFS [19]. In our study, STR (HR, 3.2; 95% CI, 1.1 to 9.4; *p* = 0.03) was associated with a recurrence, but not adjuvant radiation (HR, 2.1; 95% CI, 0.73 to 6.3; *p* = 0.17). With a bigger sample size and longer follow-up, the data on adjuvant radiation would have been different. As EEA continues to be utilized for complex tumors, assessment of outcomes should include both oncologic and quality of life (QOL) assessments. Many centers have begun to report long-term patient morbidity and evaluation or development of validated QOL instruments for a variety of tumors undergoing EEA [20,21,22,23,24].

Recurrence can be challenging to manage. If the recurrence is focal, repeated surgical resection can be done; however, if recurrence occurs with multiple intracranial metastasis or extracranial metastasis, we must consider re-radiation or chemotherapy [25]. Re-radiating the same region bears the risk of osteonecrosis and adverse effects on the nearby nerves or brainstem. Currently, chemotherapeutic drugs for advanced chordoma are tyrosine kinase inhibitors such as imatinib and immune checkpoint inhibitors such as pembrolizumab and nivolumab [26,27,28]. However, none of these drugs are approved [26]. In our cohort, one patient before 2013 and three patients after 2013 received chemotherapy after a recurrence. Imatinib, ipilimumab, or nivolumab were used for systemic metastasis, leptomeningeal dissemination, and multiple recurrences refractory to surgical resection and the previous drug.

Consequently, preventing recurrence is essential. Our result shows that patients treated with multidisciplinary therapy after 2013, GTR/NTR, and proton beam therapy were less associated with recurrences (Table 4). The same three factors were associated with less mortality (not shown). However, these results are biased by the shorter length of follow-up in patients treated after 2013 than those before 2013. This explains why multidisciplinary therapy after 2013 and proton beam therapy were not significant in Cox regression analysis (Table 5). 5-year PFS of “after 2013” (36%) was much smaller than “before 2013” (73%; Table 6). This counterintuitive result is due to the shorter follow-up period of “after 2013”. After 3-year, two patients recurred at 43 months and 44 months (7 accumulated recurrences). Furthermore, eight patients were counted out because they were not followed until 5-year. Nevertheless, these results are consistent with the past literature [6,29], which showed that EEA, GTR, and proton beam therapy were correlated with a better survival rate. Consequently, maximal safe resection followed by proton beam therapy may be the best treatment for clival chordomas.

Although most clival chordomas were resected only by EEA after 2013, certain chordomas exist where EEA is not adequate to treat the entirety of the tumor (Figure 3). We need to consider the chordoma location both in the midline clivus and lateral extent to choose an appropriate surgical approach [30]. If a chordoma involves the occipito-cervical joint (Figure 3c,d), occipito-cervical fusion is necessary to prevent postoperative instability after maximal safe resection [31]. In such cases, occipito-cervical fusion before tumor resection was reported to be better than a fusion after tumor resection [31]. In our cohort, two cases underwent an occipito-cervical fusion before EEA, and one underwent a fusion after EEA. The one who had a fusion after EEA had a minor wound complication related to the fusion surgery. None of the two cases where a fusion was performed before EEA resulted in complications. These outcomes are consistent with the previous report [31]. If the tumor extends beyond the region where EEA is limited or if it is hard to safely resect the tumor due to scar tissue, using an endoscope via a different surgical corridor (high transcervical approach [32] in our study) or open resection via craniotomy can be chosen to pursue maximal safe resection. Our series used the endoscopic transcervical approach in two chordoma cases extending as low as periodontoid space (Figure 3e–g). These two cases resulted in STR. As discussed in a previous report, it is difficult for chordomas in the upper cervical regions to achieve margin-free en bloc resection without causing significant morbidity [32]. EEA, followed by an open resection, was done in two cases. We attained the NTR in one case (Figure 3h,i) and the STR in the other case. Additional surgery by craniotomy was chosen because the tumor was firm in both cases, and there was scar tissue around the tumor and the internal carotid artery, making it challenging to resect under the endoscope. Despite the recent advancement of endoscopic surgical armamentarium, certain situations still exist where tumor resection is not feasible under the endoscope. In those cases, upfront open surgery should be considered [29]. In our cohort, three cases were treated by an open resection after 2013. In all these cases, chordomas extensively occupied the prepontine and interpeduncular cisterns displacing the brainstem (Figure 3j,k).

It is consensus that radiotherapy is recommended after complete macroscopic surgery. However, it is based on the level of evidence V and recommendation B, which means that this is based on studies without a control group, case reports, and experts’ opinions with limited benefit [33]. This is in line with the result of our study. We could not show that adjuvant radiation is associated with less recurrence (Table 5; Figure 2d). This may reflect the fact that chordomas are resistant to radiation [34]. This radioresistant makes radiation therapy difficult concerning adverse effects. Generally, a dose of about 74 Gy is recommended for chordomas [33,35,36]. This is close to the median dose of 73.5 Gy in our patients who received adjuvant proton beam therapy. Adverse radiation effects on surrounding organs at risk should be considered to give such a high dose to tumor volume. Contrary to traditional radiation by photon, radiation by a proton beam is advantageous in its ability to tailor to the tumor’s exact size and shape, potentially decreasing the adverse effects. The local control rates of conventional radiotherapy, radiosurgery, and proton beam therapy are comparable, and there is no superiority among them [29].

Intrinsic tumor factors such as genetic information should also be considered. Thanks to recent advancements in molecular biology, certain genetic factors, such as T gene gain, expression of brachyury, chromosome 1p loss, chromosome 1q gain, chromosome 9p loss, PTEN, and TP53, are found to be associated with worse outcomes [37,38,39,40]. In our series, two recurrent cases had 1p loss and 1q gain. Both recurred within two years after surgeries in outside hospitals. The only one case which did not show brachyury expression (3%) has been recurrence-free. Since we did not perform genetic analysis on most specimens, we could not assess these factors. Zenonos et al. suggest that adjuvant radiation is unnecessary if GTR is attained for clival chordomas with the best prognostic group. They defined the best prognostic group as tumors whose chromosome 1p loss is 0–15%, and 9p loss is 0–3% [40].

The comparison of outcomes between our study and previous studies is summarized in Table 7 [5,6,7,14,16,25,41,42]. 5-year PFS and OS are comparable. Although the follow-up period of our cohort after 2013 was not long enough, overall, our follow-up period is comparable to other studies.

With the introduction of endoscopy training in 2013, the number of endoscopic surgery increased, resulting in a higher rate of GTR/NTR. After 2013, the average annual number of surgical cases increased from 2.8 (2013–2016) to 5.6 (2017–2021) (*p* = 0.17). This doubling of the number of surgical cases after 2013 may reflect the refinement of the surgical expertise and learning curve in dealing with clival chordomas in the team. After 2013, more patients underwent adjuvant proton beam radiation. Theoretically, these changes should result in a better local control rate of clival chordomas. However, we need more time and cases to prove less recurrence rate.

Our study has several limitations attributed to the retrospective nature of the study, with a small number of cases and a relatively short follow-up period. Due to the small sample size (n = 58), our results should be cautiously interpreted and should not be readily generalized. In addition, due to the shorter follow-up period in the patients “after 2013”, the outcome may be different with continued follow-up. We will continue to follow the patients and re-evaluate at the appropriate time. Some of the latest MRI follow-ups were unavailable in patients treated before 2013, consequently underestimating the recurrence rate of patients from before 2013. The data on chordoma subtypes were available in only 17% (10 of 58 patients). Consequently, we did not include subtypes as a variable in the analyses. This could have impacted the analysis of our results. Poorly differentiated chordoma was reported to have a poor prognosis in a previous paper (analysis on seven patients; age range, 1 to 11 years) [43].

## 5. Conclusions

Specialized training and a multidisciplinary team improved resection rates compared to a historical cohort with an excellent morbidity profile. This further supports sub-specialty care and likely demonstrates an improved understanding of chordoma.

## Figures and Tables

**Figure 1 cancers-14-05104-f001:**
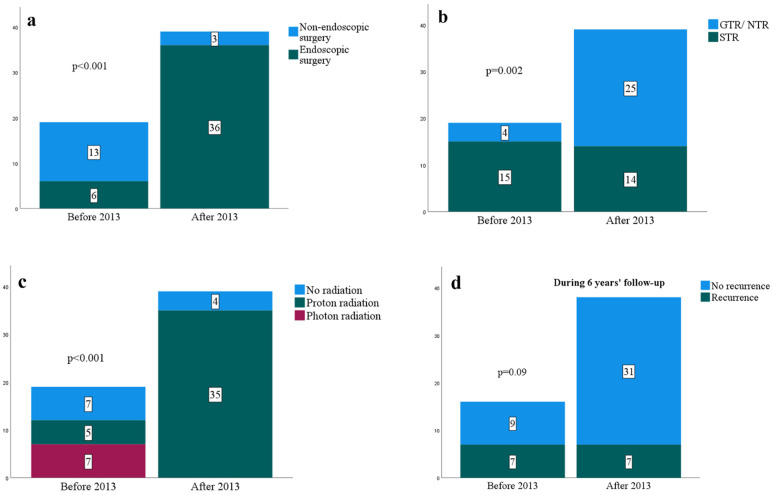
Chart bars comparing various factors before 2013 and after 2013. (**a**) Endoscopic surgeries; (**b**) Resection degree; (**c**) Adjuvant radiation status (**d**) Recurrences in the first six years.

**Figure 2 cancers-14-05104-f002:**
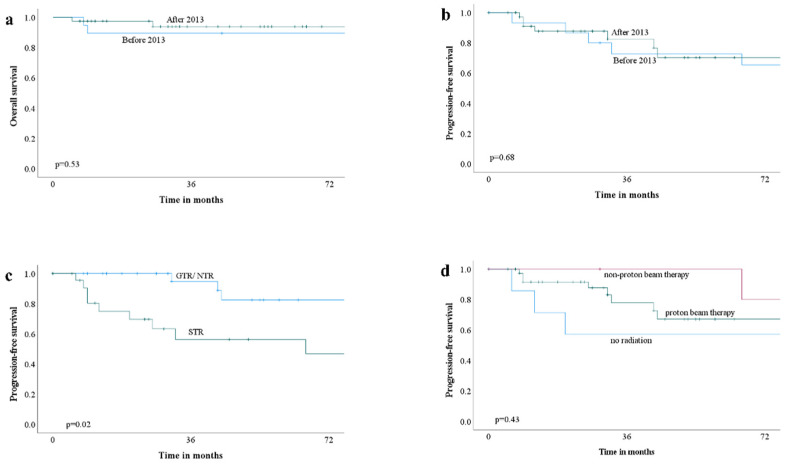
Kaplan–Meier curves of overall survival and progression-free survival (PFS). (**a**) Overall survivals of “before 2013” and “after 2013”; (**b**) PFS curves of “before 2013” and “after 2013”; (**c**) PFS curves based on the resection degree; (**d**) PFS curves based on adjuvant radiation status.

**Figure 3 cancers-14-05104-f003:**
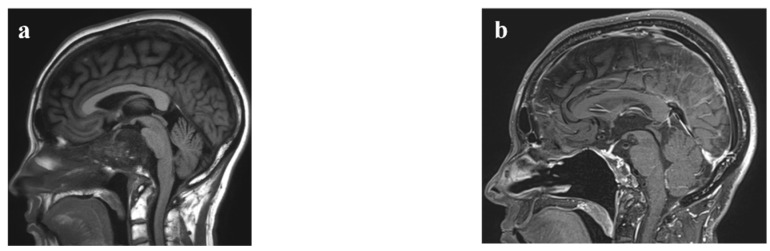
Magnetic resonance imaging (MRI) of five illustrative clival chordoma cases which required a different surgical strategy. (**a**) The first case is a holoclival chordoma (maximal diameter, 5 cm), which was resected by endoscopic endonasal approach (EEA). Preoperative T1-weighted (T1) fluid attenuated inversion recovery (FLAIR) sagittal MRI shows a clival mass displacing the pituitary gland and the brainstem; (**b**) The near total resection was attained by EEA. Enhanced T1 3D fast spin echo fat suppressed sagittal MRI in 3 years shows no recurrent tumor; (**c**) The second case is a holoclival chordoma (maximal diameter, 5 cm), which was resected by EEA after an occipito-cervical fusion (two stages). Preoperative enhanced T1 sagittal MRI shows a clival mass; (**d**) It was eroding the left occipital condyle (not shown) and an occipito-cervical fusion was done before EEA. Enhanced T1 fat suppressed sagittal MRI in 2.5 years shows no signs of recurrence; (**e**) The third case is a holoclival chordoma (maximal diameter, 8.8 cm), which was resected by endoscopic transcervical approach. Preoperative enhanced T1 fat suppressed sagittal MRI shows a clival mass extending as low as periodontoid space; (**f**) This case underwent two EEA and occipito-cervical fusion in an outside hospital. Enhanced T1 sagittal MRI shows a residual mass in the periodontoid space; (**g**) The subtotal resection was attained. T1 fast spin echo sagittal MRI in 1.5 years shows a stable residual mass; (**h**) The fourth case is a lower clival chordoma (maximal diameter, 4.3 cm), which was resected by EEA and open craniotomy. Preoperative enhanced T1 fast spin echo sagittal MRI shows an enhancing lobulated mass compression the cervicomedullary region; (**i**) EEA showed that the tumor was firm and arachnoid scarring was appreciated. One month after EEA, open tumor resection was done by right far lateral supracondylar craniotomy and C1 laminectomy (near total resection). Enhanced T1 spin echo sagittal MRI in about 8 years shows no signs of recurrence; (**j**) The last case is a holoclival chordoma (maximal diameter, 4.4 cm), which was resected by upfront open surgery. Preoperative enhanced T1 fat suppressed sagittal MRI shows an enhancing clival lobulated mass displacing the brainstem; (**k**) An open resection by left presigmoid subtemporal craniotomy with posterior petrosectomy was done (subtotal resection). T1 FLAIR sagittal MRI in 2.5 years shows a stable residual disease.

**Table 1 cancers-14-05104-t001:** Characteristics of the patients at baseline and treatment outcome.

	Total	Before 2013	After 2013	*p*
n	58	19	39	-
Age	45 ± 17	42 ± 18	46 ± 17	0.63
Male	31 (53%)	9 (47%)	22 (56%)	0.52
Body mass index	30 ± 6	27 ± 6.2	31 ± 5.9	0.78
Referred after a biopsy at an outside hospital	3	2	1	0.25
Previous treatment at an outside hospital	11	0	11	0.01
Revision surgery	10	0	10	0.02
Previous radiation at an outside hospital	0	0	1	1
Recurrent case	5	0	5	0.16
Size (cm)	3.7 ± 1.6	3.7 ± 1.5	3.6 ± 1.6	0.92
Location				-
Upper clivus	25 (43%)	9	16	
Middle clivus	14	4	10	
Lower clivus	7 (12%)	3	4	
Holoclivus	12	3	9	
Operative approach	36			-
EEA	1	6 (32%)	30 (77%)	
Endoscopic transcervical	4	0	1	
MicroEA	10	4	0	
Open craniotomy	2	7	3	
EEA, open craniotomy	2	0	2	
EEA, OC fusion	12	0	2	
EEA, OC fusion, endoscopic transcervical		0	1	
Midline mandibular osteotomy		2	0	
Degree of resection				0.002
GTR or NTR	28	3 (16%)	25 (64%)	
STR	30	16	14	
The location of residual tumor after STR				-
EEA				
Posterior to paraclival ICA, petrous apex, cavernous sinus	8	3	5	
Dorsum sella, cavernous sinus	2	1	1	
Prepontine cistern (attached to the pons)	1	-	1	
Sella, sphenoid sinus, clivus	2	1	1	
Endoscopic transcervical				
Peri-odontoid space	1	-	1	
MicroEA				
Posterior to paraclival ICA	1	1	-	
Image unavailable	3	3	-	
Open craniotomy				
Various sites (refer to the manuscript)	6	3	3	
Image unavailable	3	3	-	
EEA, open craniotomy				
Posterior suprasellar region	1	-	1	
EEA, OC fusion, endoscopic transcervical				
Prevertebral space of the craniovertebral junction	1	-	1	
Midline mandibular osteotomy				
Image unavailable	1	1	-	
New cranial nerve deficits	7	6 (32%)	1 (2.6%)	0.004
CSF leak	4	0	4	0.29
Additional surgery before radiation at an outside hospital	3	1	2	1
Adjuvant radiation				0.03
Yes	47 (81%)	12	35	
No	11	7	4	
Adjuvant radiation				<0.001
Proton beam therapy	40	5	35	
GKS and/or EBRT	7	7	0	
Recurrence	17	10	7	0.005
Death	11	9	2	<0.001
Progression-free survival	31 (IQR 13–64)	88 ± 67	29 (IQR 11–51)	-
Overall survival	54 (IQR 26–102)	142 ± 77	35 (IQR 14–57)	-

Parametric data are shown in mean ± standard deviation. Non-parametric data are shown in median with IQR. Abbreviation: CSF, cerebrospinal fluid; EBRT, external beam radiation therapy; EEA, endoscopic endonasal approach; GKS, Gamma Knife surgery; GTR, gross total resection; ICA, internal carotid artery; IQR, interquartile range; MicroEA, microscopic endonasal approach; NTR, near total resection; OC, occipito-cervical; STR, subtotal resection.

**Table 2 cancers-14-05104-t002:** The operative approach and the degree of resection based on the clival chordoma location.

	Before 2013		After 2013	
Clival Region	Approach	Degree of Resection	Approach	Degree of Resection
Upper	EEA (4), craniotomy (3), microEA (2)	GTR (1), STR (8)	EEA (14), EEA + craniotomy (1), craniotomy (1)	GTR (10), STR (6)
Middle	Craniotomy (3), microEA (1)	GTR (1), STR (3)	EEA (10)	GTR/NTR (8), STR (2)
Lower	Midline mandibular osteotomy (2), craniotomy (1)	GTR (1), STR (2)	EEA (1), EEA + craniotomy (1), EEA + OC fusion (1), EEA + OC fusion + endoscopic transcervical (1)	GTR/NTR (2), STR (2)
Holo	EEA (2), microEA (1)	STR (3)	EEA (5), craniotomy (2), EEA + OC fusion (1), endoscopic transcervical (1)	GTR/NTR (5), STR (4)

Abbreviation: EEA, endoscopic endonasal approach; GTR, gross total resection; microEA, microscopic endonasal approach; NTR, near-total resection; OC, occipito-cervical; STR, subtotal resection.

**Table 3 cancers-14-05104-t003:** The result of binary logistic regression analysis on postoperative new cranial nerve deficits.

	Cranial Nerve Deficits
Time	
After 2013	Reference
Before 2013	17 (1.9–160), 0.001
Age	0.99 (0.94–1.0), 0.59
Sex	
Male	Reference
Female	0.84 (0.17–4.2), 0.84
Body mass index	0.85 (0.71–1.0), 0.059
Revision surgery	
No	Reference
Yes	0.0 (0.0- ), 1
Recurrent case	
No	Reference
Yes	0.0 (0.0-), 1
Size	1.4 (0.86–2.1), 0.19
Location	
Upper	Reference
Middle	0.56 (0.053–6.0), 0.64
Lower	2.9 (0.38–22), 0.30
Whole	0.67 (0.062–7.2), 0.74
Degree of resection	
GTR/NTR	Reference
STR	2.8 (0.50–16), 0.24

The odds ratio, its 95% confidence interval in the parenthesis, and *p*-value are written. Abbreviation: GTR, gross total resection; NTR, near total resection; STR, subtotal resection.

**Table 4 cancers-14-05104-t004:** The result of binary logistic regression analysis on a recurrence.

	Univariate	Multivariate
Time		
After 2013	Reference	Reference
Before 2013	5.5 (1.6–19), 0.007	3.8 (0.71–21), 0.12
Age	1.0 (0.98–1.0), 0.46	-
Sex		-
Male	Reference	
Female	0.52 (0.16–1.7), 0.27	
Body mass index	0.94 (0.85–1.0), 0.20	-
Previous treatment	0.52 (0.097–2.7), 0.44	-
Revision surgery		-
No	Reference	
Yes	0.61 (0.11–3.3), 0.57	
Recurrent case		-
No	Reference	
Yes	0.55 (0.057–5.3), 0.60	
Size	1.2 (0.83–1.7), 0.36	-
Location		-
Upper	Reference	
Middle	1.6 (0.39–6.6), 0.51	
Lower	1.9 (0.34–11), 0.46	
Whole	0.57 (0.098–3.3), 0.53	
Degree of resection		
GTR/NTR	Reference	Reference
STR	3.8 (1.1–13), 0.03	2.2 (0.53–8.8), 0.28
Adjuvant radiation		
None	Reference	Reference
Proton beam therapy	0.13 (0.026–0.61), 0.01	0.30 (0.048–1.9), 0.20
GKS and/or EBRT	0.38 (0.049–2.9), 0.35	0.21 (0.022–1.9), 0.17

The odds ratio, its 95% confidence interval in the parenthesis, and *p*-value are written. Abbreviation: EBRT, external beam radiation therapy; GKS, Gamma Knife surgery; GTR, gross total resection; NTR, near total resection.

**Table 5 cancers-14-05104-t005:** The result of Cox regression analysis on a recurrence.

Factor	Hazard Ratio (95% Confidence Interval), *p*
Time	
After 2013	Reference
Before 2013	1.3 (0.42–3.8), 0.68
Age	1.0 (0.99–1.1), 0.15
Sex	
Male	Reference
Female	0.84 (0.30–2.3), 0.73
Body mass index	0.98 (0.90–1.1), 0.73
Previous treatment	2.0 (0.43–9.6), 0.38
Revision surgery	2.5 (0.53–12), 0.25
Size	1.2 (0.94–1.6), 0.14
Location	
Upper	Reference
Middle	1.2 (0.34–4.0), 0.80
Lower	2.1 (0.53–8.6), 0.29
Whole	0.63 (0.13–3.0), 0.57
Degree of resection	
GTR/NTR	Reference
STR	3.2 (1.1–9.4), 0.033
Adjuvant radiation	
None	Reference
Proton beam therapy	0.51 (0.16–1.7), 0.27
GKS and/or EBRT	0.40 (0.094–1.7), 0.21
Adjuvant radiation	
Yes	Reference
No	2.1 (0.73–6.3), 0.17

Abbreviation: EBRT, external beam radiation therapy; GKS, Gamma Knife surgery; GTR, gross total resection; NTR, near total resection.

**Table 6 cancers-14-05104-t006:** The rate of progression-free survival (PFS) and overall survival (OS) between the two time periods.

	**1-Year**		**3-Year**		**5-Year**		**10-Year**	
	**PFS**	**OS**	**PFS**	**OS**	**PFS**	**OS**	**PFS**	**OS**
Before 2013	94% 15/16	89% 17/19	73% 11/15	89% 17/19	73% 11/15	89% 16/18	40% 6/15	72% 13/18
After 2013	87% 27/31	97% 31/32	74% 14/19	90% 19/21	36% 4/11	83% 10/12	-	-
Overall	89% 42/47	94% 48/51	74% 25/34	90% 36/40	58% 15/26	87% 26/30	-	-

**Table 7 cancers-14-05104-t007:** Comparison with past literature. PFS and OS are comparable to other studies.

**Authors, Year**	**N**	**GTR/NTR**	**Postop CND**	**CSF Leak (%)**	**Adjuvant RT; PBT**	**Recurrence (%)**	**PFS/OS (%)**	**Follow-Up (Months)**
Chibbaro et al., 2014 [16]	54 patients (58 EEA)	44 (81%)	ND	4 (7%)	54 (100%); 54 (100%)	4 (11%)	ND/ND	Mean, 34
Forander et al., 2017 [41]	22 (EEA 11)	3 (14%)	4 (19%)	2 (10%)	ND; 3 (14%)	ND	5y: ND/82	Median, 80
Jagersberg et al., 2017 [25]	13 (TSS 7)	2 (15%)	4 (31%)	2 (15%)	13 (100%); 11 (85%)	7 (54%)	5y: 53/83	Mean, 78; median, 64
Wang et al., 2017 [14]	238 (TSS 51)	157 (66%)	ND	9 (4%)	ND	82 (46%)	5y: 45/76	Mean, 44
Zoli et al., 2018 [7]	65 patients (EEA 80 cases)	47 (59%)	7 (9%)	2 (3%)	63 (97%); 60 (92%)	24 (37%)	5y: ND/77	Mean, 52; median, 48
La Corte et al., 2021 [42]	59 patients (EEA 26)	12 (20%) *	7 (12%)	9 (15%)	44 (75%); 19 (32%)	37 (63%)	Mean: 65 months/83 months	Mean, 83
Zweckberger et al., 2020 [5]	50 patients; 70 cases (TSS 24)	12 (24%)	ND	9 (13%)	35 (50%); 2 (3%)	ND	5y: 45/ND	ND
Cavallo et al., 2020 [6]	167 patients; 182 cases (EEA 151)	93 (51%)	ND	9 (5%)	141 (78%); 115 (63%)	49 (27%)	5y: 62/74	Mean, 62
Our study	58 patients; 58 cases (EEA 36)	28 (48%)	7 (12%)	4 (7%)	47 (81%); 40 (69%)	17 (29%)	5y: 58/87	Mean, 75; median, 54

* Only GTR. Abbreviation: CND, new-onset or worsened cranial nerve deficits; EEA, endoscopic endonasal approach; GTR, gross total resection; ND, not described; OS, overall survival; PBT, proton beam therapy; PFS, progression-free survival; RT, radiation therapy; TSS, trans-sphenoidal surgery.

## Data Availability

The data presented in this study are available on request from the corresponding author.

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
