# Peer review of "Mayo Clinic Clival Chordoma Case Series: Impact of Endoscopic Training on Clinical Care"

_cancers, 2022, doi:10.3390/cancers14205104_

Round 1

Reviewer 1 Report

Thank you to the authors for your work on this interesting manuscript and important topic. Most otolaryngologists do encounter patients with Clival Chordoma and are certainly aware of the impact and on our limitations in treating the symptom effectively.

1. Great job with describing all statistical tests in the biostatistics paragraph.    The sample size is small and you did address on normality of the data. 

The following issues will need to be addressed:

This is certainly an interesting area of study.  I am looking forward to the outcomes of rigorous prospective investigation on this topic.  While this study is very interesting, many limitations exist in this retrospective design and were not noted by the authors.  Please discuss limitations in the Discussion section.  

 The very small sample size makes it difficult to begin to generalize the outcomes. It is important that that the authors state that readers should avoid this for this reason.

Author Response

To the comment “Thank you to the authors for your work on this interesting manuscript and important topic. Most otolaryngologists do encounter patients with Clival Chordoma and are certainly aware of the impact and on our limitations in treating the symptom effectively. 1. Great job with describing all statistical tests in the biostatistics paragraph. The sample size is small and you did address on normality of the data.”,

            Thank you for the encouraging comments.

To the comment “This is certainly an interesting area of study.  I am looking forward to the outcomes of rigorous prospective investigation on this topic.  While this study is very interesting, many limitations exist in this retrospective design and were not noted by the authors.  Please discuss limitations in the Discussion section. The very small sample size makes it difficult to begin to generalize the outcomes. It is important that that the authors state that readers should avoid this for this reason.”,

We corrected the manuscript as follows.

Line 408-412: “,,, Due to the small sample size (n=58), our results should be cautiously interpreted and should not be readily generalized. In addition, due to the shorter follow-up period in the patients “after 2013”, the outcome may be different with continued follow-up. We will continue to follow the patients and re-evaluate at the appropriate time. ,,,”

Reviewer 2 Report

The authors (AA) performed a retrospective study on clival chordoma treated at their institution in a frame time of around 30 years. They concluded that “a specialized multidisciplinary team improved the resection rate compared to a historical cohort with an excellent morbidity profile”. Although the paper may represent another contribution in the chordoma literature, there are some concerns that need to be addressed before.  

-        The authors should elucidate which craniotomy approaches have been used when they refer to “open craniotomy”

-        The authors should also define the lateral extent of chordoma rather the only anatomical midline clival anatomical location, since this could impact on the relevance of the choice of the endoscopic endonasal approach vs craniotomy and the presence of any residuals[1]

-        The authors should report on the location of the tumor residual in the STR group

-        On the authors’ multivariable analysis, they found some not new findings related to chordoma recurrence, specifically that radiation therapy and entity of resection represent the most important prognostic factors impacting the progression survival. The authors should cite the following systematic review on prognostic factors and discuss it in light of the authors findings[2]

-        Page 5, line 130, The authors should correct the sentence punctuation. The presence of any pathology data in only 17% of patients is scarce. This represents a limitation in the interpretation of the authors results. The presence of any de-differentiated chordoma should be clearly reported, since this could impact the patient outcome OS and PFS[3]

-        The AA should clearly state the median/mean follow-up of their whole series in the abstract and results section

-        The AA only mention the use of fascia lata when reporting on endoscopic skull base closure. Did the authors use a vascularized pedicled flaps or some other techniques (i.e. gasket seal?). I think that a brief comment on that would be worth, since the raise of CSF leaks in the last decade because of the use of EEA[4].

-        The authors should also comment on the impact of the surgical expertise and learning curve in dealing such challenging tumors in a so extended time period. A previous paper looked on the team expertise in skull base surgery that significantly impacted on PFS of chordoma patients; since the present case series has one of the longest reported follow-up and clearly state how the role of endoscopy in different time eras affected the surgical outcomes of chordoma patient, it should also be included in the discussion section and table 6[5].

References

1.             La Corte, E.; Broggi, M.; Bosio, L.; Danesi, G.; Ferroli, P. Tailored surgical strategy in clival chordomas: an extraordinary selection bias that limits approach comparison. J. Neurosurg. Sci. 201862, 519–521.

2.             Zou, M.X.; Lv, G.H.; Zhang, Q.S.; Wang, S.F.; Li, J.; Wang, X. Bin Prognostic Factors in Skull Base Chordoma: A Systematic Literature Review and Meta-Analysis. World Neurosurg. 2018109, 307–327.

3.             Hasselblatt, M.; Thomas, C.; Hovestadt, V.; Schrimpf, D.; Johann, P.; Bens, S.; Oyen, F.; Peetz-Dienhart, S.; Crede, Y.; Wefers, A.; et al. Poorly differentiated chordoma with SMARCB1/INI1 loss: a distinct molecular entity with dismal prognosis. Acta Neuropathol. 2016132, 149–151.

4.             Garcia-Navarro, V.; Anand, V.K.; Schwartz, T.H. Gasket seal closure for extended endonasal endoscopic skull base surgery: efficacy in a large case series. World Neurosurg. 201380, 563–8.

5.             La Corte, E.; Broggi, M.; Raggi, A.; Schiavolin, S.; Acerbi, F.; Danesi, G.; Farinotti, M.; Felisati, G.; Maccari, A.; Pollo, B.; et al. Peri-operative prognostic factors for primary skull base chordomas: results from a single-center cohort. Acta Neurochir. (Wien). 2020.

Author Response

To the comment “The authors (AA) performed a retrospective study on clival chordoma treated at their institution in a frame time of around 30 years. They concluded that “a specialized multidisciplinary team improved the resection rate compared to a historical cohort with an excellent morbidity profile”. Although the paper may represent another contribution in the chordoma literature, there are some concerns that need to be addressed before.”,

            Thank you for the valuable comments. We corrected our manuscript based on your comments as follows.   

To the comment “The authors should elucidate which craniotomy approaches have been used when they refer to “open craniotomy””,

            We corrected the manuscript as follows.

            Line 126-130, “Open craniotomy was used in 12 patients with or without EEA. Subtemporal craniotomy was most commonly utilized (four patients) followed by frontotemporal craniotomy (three patients), suboccipital craniotomy (two patients), temporal occipital craniotomy (two patients), and far lateral supracondylar craniotomy (one patient).”

To the comment “The authors should also define the lateral extent of chordoma rather the only anatomical midline clival anatomical location, since this could impact on the relevance of the choice of the endoscopic endonasal approach vs craniotomy and the presence of any residuals[1]”,

            We corrected the manuscript and made a new table as follows.

            Line 68-70, “As for lateral extension, the upper clivus included the cavernous sinus. The middle clivus included the petrous bone (the petroclival fissure). And the lower clivus included the jugular foramen and the occipital condyle.”

            Line 133-136, “The operative approach and the resection degree based on the four clival chordoma classification is summarized in Table 2. After 2013, the middle clival chordoma cases were all resected by EEA. And GTR was accomplished in 80%, which was higher than the other clival regions.”

            Line 311-313, “We need to consider the chordoma location both in the midline clivus and lateral extent to choose an appropriate surgical approach.{La Corte, 2018 #131}”

Table 2. The operative approach and the degree of resection based on the clival chordoma location.

Before 2013

After 2013

Clival region

Approach

Degree of resection

Approach

Degree of resection

Upper

EEA (4), craniotomy (3), microEA (2)

GTR (1), STR (8)

EEA (14), EEA + craniotomy (1), craniotomy (1)

GTR (10), STR (6)

Middle

Craniotomy (3), microEA (1) 

GTR (1), STR (3)

EEA (10)

GTR/ NTR (8), STR (2)

Lower

Midline mandibular osteotomy (2), craniotomy (1)

GTR (1), STR (2)

EEA (1), EEA + craniotomy (1), EEA + OC fusion (1), EEA + OC fusion + endoscopic transcervical (1)

GTR/ NTR (2), STR (2)

Holo

EEA (2), microEA (1)

STR (3)

EEA (5), craniotomy (2), EEA + OC fusion (1), endoscopic transcervical (1)

GTR/ NTR (5), STR (4)

Abbreviation: EEA, endoscopic endonasal approach; GTR, gross total resection; microEA, microscopic endonasal approach; NTR, near-total resection; OC, occipito-cervical; STR, subtotal resection.

To the comment “The authors should report on the location of the tumor residual in the STR group”,

            We added a row in Table 1 as follows.

The location of residual tumor after STR

EEA

 Posterior to paraclival ICA, petrous apex, cavernous sinus

 Dorsum sella, cavernous sinus

 Prepontine cistern (attached to the pons)

 Sella, sphenoid sinus, clivus

Endoscopic transcervical

 Peri-odontoid space

MicroEA

 Posterior to paraclival ICA

 Image unavailable

Open craniotomy

 Various sites (refer to the manuscript)

 Image unavailable

EEA, open craniotomy

 Posterior suprasellar region

EEA, OC fusion, endoscopic transcervical

 Prevertebral space of the craniovertebral junction

Midline mandibular osteotomy

 Image unavailable

8

2

1

2

1

1

3

6

3

1

1

1

3

1

-

1

-

1

3

3

3

-

-

1

5

1

1

1

1

-

-

3

-

1

1

-

-

            We corrected the manuscript as follows.

            Line 136-145, “The location of the tumor residual in the STR group is summarized in Table 1. After EEA, the most common location was posterior to the paraclival internal carotid artery, the petrous apex, and the cavernous sinus. After open craniotomy, the residual tumor location varied depending on the craniotomy sites. After subtemporal craniotomy, the residuals ranged from the prepontine cistern (one patient), the cavernous sinus/ petrous apex/ cerebellopontine angle (one patient), the sella/ suprasellar regions (one patient). After temporal occipital craniotomy, all the residuals were in the prepontine cistern (two patients). After suboccipital craniotomy, the residual was in the cavernous sinus to medial to the petrous segment internal carotid artery (one patient).”

To the comment “On the authors’ multivariable analysis, they found some not new findings related to chordoma recurrence, specifically that radiation therapy and entity of resection represent the most important prognostic factors impacting the progression survival. The authors should cite the following systematic review on prognostic factors and discuss it in light of the authors findings[2]”,

            We corrected the manuscript as follows.

            Line 275-280, “A meta-analysis on prognostic factors in skull base chordoma showed that STR (pooled HR, 2.0; 95% CI, 1.5 to 2.6) and adjuvant radiation (pooled HR, 0.30; 95% CI, 0.16 to 0.56) were associated with PFS.[1] In our study, STR (HR, 3.2; 95% CI, 1.1 to 9.4; P=0.03) was associated with a recurrence, but not adjuvant radiation (HR, 2.1; 95% CI, 0.73 to 6.3; P=0.17). With a bigger sample size and longer follow-up, the data on adjuvant radiation would have been different.”

To the comment “Page 5, line 130, The authors should correct the sentence punctuation. The presence of any pathology data in only 17% of patients is scarce. This represents a limitation in the interpretation of the authors results. The presence of any de-differentiated chordoma should be clearly reported, since this could impact the patient outcome OS and PFS[3]”,

We corrected the manuscript as follows.

            Line 154, “No patients had de-differentiated chordomas.”

            Line 413-417, “The data on chordoma subtypes were available in only 17% (10 of 58 patients). Consequently, we did not include subtypes as a variable in the analyses. This could have impacted the analysis of our results. Poorly differentiated chordoma was reported to have a poor prognosis in a previous paper (analysis on seven patients; age range, 1 to 11 years).”

To the comment “The AA should clearly state the median/mean follow-up of their whole series in the abstract and results section”,

            We corrected the manuscript as follows.

            Line 28-30, “For cases before 2013, 10 patients (53%) recurred during the median follow-up of 144 months (mean, 142 months), whereas for cases after 2013, seven patients (18%) recurred with a median follow-up of 35 months (mean, 42 months).”

            Line 204, “The mean follow-up period was 142 months (median, 144 months).”

            Line 206-207, “The mean follow-up period was 42 months (median, 35 months).”

To the comment “The AA only mention the use of fascia lata when reporting on endoscopic skull base closure. Did the authors use a vascularized pedicled flaps or some other techniques (i.e. gasket seal?). I think that a brief comment on that would be worth, since the raise of CSF leaks in the last decade because of the use of EEA[4].”,

            We corrected the manuscript as follows.

            Line 174-176, “The nasoseptal flap was used in 24 patients (five of seven no CSF leaks; six of six low-flow CSF leaks; 13 of 17 high-flow CSF leaks). The gasket seal technique {Garcia-Navarro, 2013 #136} was not used in any cases.”

            Line 255-259, “Furthermore, the gasket seal technique where fascia lata is stabilized by a rigid buttress may reinforce the water tightness of the skull base defect. In one study, where the authors performed the gasket seal technique on 46 patients comprised of craniopharyngioma, meningioma, and pituitary adenoma, only two patients (4%) had a postoperative CSF leak.{Garcia-Navarro, 2013 #136}”

To the comment “The authors should also comment on the impact of the surgical expertise and learning curve in dealing such challenging tumors in a so extended time period. A previous paper looked on the team expertise in skull base surgery that significantly impacted on PFS of chordoma patients; since the present case series has one of the longest reported follow-up and clearly state how the role of endoscopy in different time eras affected the surgical outcomes of chordoma patient, it should also be included in the discussion section and table 6[5].”,

            We corrected the manuscript as follows.

            Line 399-402, “After 2013, the average annual number of surgical cases increased from 2.8 (2013-2016) to 5.6 (2017-2021) (P=0.17). This doubling of the number of surgical cases after 2013 may reflect the refinement of the surgical expertise and learning curve in dealing with clival chordomas in the team.”

            And we included the paper [5] in table 6.

References

  1. La Corte, E.; Broggi, M.; Bosio, L.; Danesi, G.; Ferroli, P. Tailored surgical strategy in clival chordomas: an extraordinary selection bias that limits approach comparison. J. Neurosurg. Sci. 2018, 62, 519–521.
  2. Zou, M.X.; Lv, G.H.; Zhang, Q.S.; Wang, S.F.; Li, J.; Wang, X. Bin Prognostic Factors in Skull Base Chordoma: A Systematic Literature Review and Meta-Analysis. World Neurosurg. 2018, 109, 307–327.
  3. Hasselblatt, M.; Thomas, C.; Hovestadt, V.; Schrimpf, D.; Johann, P.; Bens, S.; Oyen, F.; Peetz-Dienhart, S.; Crede, Y.; Wefers, A.; et al. Poorly differentiated chordoma with SMARCB1/INI1 loss: a distinct molecular entity with dismal prognosis. Acta Neuropathol. 2016, 132, 149–151.
  4. Garcia-Navarro, V.; Anand, V.K.; Schwartz, T.H. Gasket seal closure for extended endonasal endoscopic skull base surgery: efficacy in a large case series. World Neurosurg. 2013, 80, 563–8.

5.             La Corte, E.; Broggi, M.; Raggi, A.; Schiavolin, S.; Acerbi, F.; Danesi, G.; Farinotti, M.; Felisati, G.; Maccari, A.; Pollo, B.; et al. Peri-operative prognostic factors for primary skull base chordomas: results from a single-center cohort. Acta Neurochir. (Wien). 2020.
